# A cross-sectional evaluation of five warfarin anticoagulation services in Uganda and South Africa

**Jerome Roy Semakula**[1☺], **Johannes P. Mouton**[2☺], **Andrea Jorgensen**[3], **Claire Hutchinson**[4], **Shaazia Allie**[2], **Lynn Semakula**[1], **Neil French**[4], **Mohammed Lamorde**[1], **Cheng-Hock Toh**[5], **Marc Blockman**[2], **Christine Sekaggya-Wiltshire**[1], **Catriona Waitt**[1,4], **Munir Pirmohamed**[6], **Karen Cohen**[2]*

1 Infectious Diseases Institute, College of Health Sciences Makerere University, Kampala, Uganda, 2 Division of Clinical Pharmacology, Department of Medicine, University of Cape Town, Cape Town, South Africa, 3 Department of Biostatistics, Institute of Translational Medicine, University of Liverpool, Liverpool, United Kingdom, 4 Department of Molecular and Clinical Pharmacology, Institute of Translational Medicine, University of Liverpool, Liverpool, United Kingdom, 5 Institute of Infection and Global Health, University of Liverpool, Liverpool, United Kingdom, 6 Wolfson Centre for Personalised Medicine, Institute of Translational Medicine, University of Liverpool, Liverpool, United Kingdom

☺ These authors contributed equally to this work.
* karen.cohen@uct.ac.za

**Data Availability Statement:** All relevant data are within the manuscript and Supporting Information files.

## Abstract

### Introduction

Warfarin is the most commonly prescribed oral anticoagulant in sub-Saharan Africa and requires ongoing monitoring. The burden of both infectious diseases and non-communicable diseases is high and medicines used to treat comorbidities may interact with warfarin. We describe service provision, patient characteristics, and anticoagulation control at selected anticoagulation clinics in Uganda and South Africa.

### Methods

We evaluated two outpatient anticoagulation services in Kampala, Uganda and three in Cape Town, South Africa between 1 January and 31 July 2018. We collected information from key staff members about the clinics' service provision and extracted demographic and clinical data from a sample of patients' clinic records. We calculated time in therapeutic range (TTR) over the most recent 3-month period using the Rosendaal interpolation method.

### Results

We included three tertiary level, one secondary level and one primary level anticoagulation service, seeing between 30 and 800 patients per month. Care was rendered by nurses, medical officers, and specialists. All healthcare facilities had on-site pharmacies; laboratory INR testing was off-site at two. Three clinics used warfarin dose-adjustment protocols; these were not validated for local use. We reviewed 229 patient clinical records. Most common indications for warfarin were venous thrombo-embolism in 112/229 (49%), atrial

**Funding:** This research was commissioned by the National Institute for Health Research (NIHR) Global Health Research Group on Warfarin anticoagulation in patients with cardiovascular disease in sub-Saharan Africa [ref: 16/137/101] using UK aid from the UK Government. The views expressed in this publication are those of the author(s) and not necessarily those of the NIHR or the Department of Health and Social Care.

**Competing interests:** The authors have declared that no competing interests exist.

fibrillation in 74/229 (32%) and valvular heart disease in 30/229 (13%). Patients were generally followed up monthly. HIV prevalence was 20% and 5% at Ugandan and South African clinics respectively. Cardiovascular comorbidity predominated. Furosemide, paracetamol, enalapril, simvastatin, and tramadol were the most common concomitant drugs. Anticoagulation control was poor at all included clinics with median TTR of 41% (interquartile range 14% to 69%).

## Conclusions

TTR was suboptimal at all included sites, despite frequent patient follow-up. Strategies to improve INR control in sub-Saharan patients taking warfarin are needed. Locally validated warfarin dosing algorithms in Uganda and South Africa may improve INR control.

## Introduction

Warfarin is the most commonly prescribed oral anticoagulant in resource-limited settings, where it is used for the treatment and prophylaxis of venous thromboembolism, and for the prevention of embolic strokes in patients with atrial fibrillation (AF) and valvular heart disease. Patients receiving warfarin require ongoing monitoring of their International Normalised Ratio (INR) which needs to be kept in a defined therapeutic range determined by the indication for therapy. Under-dosing (leading to a sub-therapeutic INR) and overdosing (leading to a supra-therapeutic INR) places individuals at risk of thrombosis and bleeding, respectively. INR control in patients in sub-Saharan Africa is often poor [1–3] which may result in potentially preventable morbidity [4] and mortality [5]. The burden of HIV and tuberculosis (TB) is high in sub-Saharan Africa [6]. Patients on warfarin who are HIV positive and/or have TB require multiple concomitant medicines, which may interact with warfarin and complicate warfarin dosing and dose adjustment [7].

In high income countries, warfarin dosing is frequently guided by validated dosing algorithms, some of which include genotyping for polymorphisms important for warfarin metabolism [8]. Little work has been done on validated algorithms to guide dosing in sub-Saharan Africa. Direct oral anticoagulants (DOACs) are widely accessible in high-income settings, but higher costs limit their use in resource-limited settings. Warfarin is therefore likely to continue to be used extensively, and strategies to improve anticoagulation control on warfarin are important.

Our aim was to evaluate the quality of anticoagulation services in Uganda and South Africa and to describe the clinical and demographic characteristics of patients receiving warfarin.

## Materials and methods

### Study design and sites

We conducted the audit between 1 January and 31 July 2018. We included two outpatient clinics in Kampala, Uganda: at Mulago National Referral Hospital, and at the Uganda Heart Institute (UHI), which is a specialized cardiac centre. In South Africa, we included clinics providing outpatient anticoagulation care at Groote Schuur Hospital, Tygerberg Hospital, and Gugulethu Community Health Centre. All audited clinics were located in an urban setting. In Uganda, both clinics were located in referral hospitals where the vast majority of patients receiving warfarin anticoagulation in the country are managed. The clinics in Cape Town, South Africa, represented primary, secondary and tertiary levels of care.

We collected information from key members of clinic staff including doctors, nurses, laboratory and pharmacy staff using a structured data capture sheet. Data collected included availability and cost of anticoagulation drugs, and INR measurement and availability and use of dose adjustment protocols (see S1 Data).

We reviewed a sample of patients' clinic records. The method of sampling clinic records varied between clinic sites because of difference in systems for record storage and documentation of clinic attendance. We aimed to minimise sampling bias. However, none of the included anticoagulation clinics had a readily available sampling frame that included all the patients in care at that clinic. We therefore could not draw a truly random sample of patients in care at any of the clinics. We extracted demographic data and clinical data including indication for anticoagulation, comorbidities, concomitant medications and INR results over the six months prior to the most recent clinic visit (see S1 Data)

We did not perform a formal sample size calculation as this was a descriptive audit of service provision, and as we did not want to test a hypothesis or estimate a parameter with a certain level of precision. We aimed to review at least 200 clinical records as this was considered feasible.

## Data analysis

Qualitative data were summarised using tables with descriptive text, and quantitative data expressed as summary statistics–median and interquartile range (IQR). We calculated time in therapeutic range (TTR) over the most recent three months using the Rosendaal interpolation method [9]. We did not place any restriction on the maximum interval between INR measurements when applying the method.

We identified concomitant drugs that could interact with warfarin using Stockley's Drug Interactions [10].

We used Stata 15.1 (StataCorp, College Station, Texas, USA) and R software package 3.5.3 for statistical analysis and figure generation.

## Ethical approvals

The Joint Clinical Research Centre and Uganda National Council for Science and Technology in Uganda (HS 179ES) and the Human Research Ethics Committee at the Faculty of Health Sciences of the University of Cape Town in South Africa (585/2017) gave ethics approval. We obtained institutional clearance for the audited clinics. All staff providing information in Uganda provided written informed consent prior to participation. In South Africa, the requirement for written informed consent to be obtained from staff providing information was waived by the ethics committee.

## Results

Anticoagulation services included in this audit are described in Table 1. The number of patients receiving warfarin ranged from 30 to 800 patients per month. Specialists or medical doctors were present in all clinics except for the secondary level service in South Africa, which was staffed by nurses. All facilities had an onsite pharmacy dispensing warfarin. At the Ugandan clinics, warfarin stock outs occurred frequently.

INR turnaround time varied (see Table 1). At health facilities with onsite INR testing, results were available on the same day. However, for the tertiary hospital in Uganda and the primary health centre in South Africa, which did not have onsite INR testing services, INR results were returned the next day, requiring two consecutive visits for the patient. Only one site (Uganda Heart Institute) had point of care INR testing available.

**Table 1. Characteristics of the five outpatient anticoagulant services in Uganda and South Africa.**

| | UGANDA | | SOUTH AFRICA | | |
|---|---|---|---|---|---|
| **Facility name** | **Mulago National Referral Hospital** | **Uganda Heart Institute** | **Groote Schuur Hospital** | **Tygerberg Hospital** | **Gugulethu Community Health Centre** |
| **Level of care of clinic/s providing warfarin** | Tertiary Hospital | Tertiary Cardiac Centre | Secondary | Tertiary | Primary |
| **Service providers** | Specialist; Nurse; Medical Officer | Specialist; Nurse; Medical Officer | Nurse | Specialist; Medical Officer | Nurse; Medical Officer |
| **Clinic days per week** | 1 | 5 | 4 | 5* | 1 |
| **Patients per month** | 30 | 400 | 800 | 105 | 113 |
| **Onsite pharmacy** | Yes | Yes | Yes | Yes | Yes |
| **DOACs available** | No | Yes (rivaroxaban) | No | No | No |
| **INR turnaround time (days)** | 1** | 0 *** | 0 | 0 to 1 | 2** |
| **INR testing cost to patient per visit (GBP)** | GBP 4 | GBP 4 (laboratory INR) GBP 1 (point of care test) | No cost to patient | No cost to patient | No cost to patient |
| **Frequency of follow-up for clinically stable patients** | Monthly | Monthly | 4- to 6-weekly | Monthly | Monthly |
| **Standard protocol used for dose adjustment** | No | Yes | Yes | No | Yes |

GBP-British Pound Sterling

*There is no dedicated INR clinic. Patients on warfarin are distributed among the specialty clinics depending on where warfarin was initiated and the indication.

**No onsite INR testing services available

***Point-of-care tests available

At the Ugandan clinics, patients paid approximately £4 for an INR test; occasionally this fee was subsidised where point of care testing was available. Patients at both Ugandan clinics had to purchase warfarin themselves when there were drug stock-outs at the clinic; at a cost of between £3 and £6 for a month's supply of warfarin depending on warfarin dose. At the South African clinics, patients did not pay for INR testing or for warfarin.

Three of the included clinics used warfarin dose adjustment protocols. The dose adjustment protocol used at the Uganda Heart institute was based on the Modified Henry Ford warfarin maintenance dosing algorithm [11]. The dosing adjustment schedules used in the South African clinics were not referenced. To our knowledge these protocols had not been validated for the clinic population.

We reviewed the clinic records of 229 patients: 68 from Uganda Heart Institute, 32 from Mulago National Referral Hospital haematology clinic, 48 from Groote Schuur Hospital, 47 from Tygerberg Hospital, and 34 from Gugulethu Community Health Centre. The three most common indications for warfarin treatment were venous thrombo-embolism (49%), atrial fibrillation (32%) and valvular heart disease (including mechanical heart valves) (13%). Characteristics of included patients are summarised in Table 2. We found that 126 patients (55%) had at least one concomitant medication which could potentially interact with warfarin [10]. There were 12 patients (5.2%) on efavirenz and one patient (0.43%) on lopinavir-ritonavir.

TTR was suboptimal at all included sites (see Fig 1). Median TTR overall was 41% (interquartile range 14% to 69%).

## Discussion

This evaluation, which included outpatient clinics managing patients on warfarin at primary, secondary and tertiary level, found that TTR was suboptimal at all included sites. This occurred despite regular INR monitoring.

**Table 2. Characteristics of patients attending five outpatient anticoagulation services in Uganda and South Africa.**

| | Uganda (n = 100) | South Africa (n = 129) | Overall (n = 229) |
|---|---|---|---|
| Age in years (median [IQR]) | 57 [43 to 67] | 56 [42 to 65] | 56 [43 to 66] |
| Female (n, (%)) | 69 (69%) | 87 (67%) | 156 (68%) |
| Indication for warfarin (n, (%))* | VTE: 63 (63%) | VTE: 49 (38%) | VTE: 112 (49%) |
| | AF: 36 (36%) | AF: 38 (29%) | AF: 74 (32%) |
| | VHD: 1 (1%) | VHD: 29 (22%) | VHD: 30 (13%) |
| | Other: 0 (0%)** | Other: 17 (13%)** | Other: 17 (7%)** |
| Number of INR tests in 6 months (median [IQR]) | 3 [3 to 4] | 5 [3 to 6] | 4[3 to 6] |
| HIV-positive (n, (%)) | 20 (20%) | 7 (5%) | 27 (12%) |
| Current tuberculosis (n, (%)) | 1 (1%) | 1 (1%) | 2 (1%) |
| Five most common non-communicable comorbidities (n, (%)) | HPT: 29 (29%) | HPT: 71 (55%) | HPT: 100 (44%) |
| | HF: 10 (10%) | DM: 21 (16%) | DM: 25 (11%) |
| | HHD: 8 (8%) | IHD: 20 (16%) | IHD: 23 (10%) |
| | DCMO: 6 (6%) | Dyslipidaemia: 19 (15%) | HF: 20 (9%) |
| | DM: 4 (4%) | HF: 10 (8%) | Dyslipidaemia: 19 (7%) |
| Ten most commonly used concomitant medicines (n, (%)) | Furosemide: 43 (43%) | Paracetamol: 59 (46%) | Furosemide: 90 (39%) |
| | Spironolactone: 21 (21%) | Enalapril: 50 (39%) | Paracetamol: 59 (26%) |
| | Bisoprolol: 19 (19%) | Simvastatin: 49 (38%) | Enalapril: 52 (23%) |
| | Digoxin: 16 (16%) | Furosemide: 47 (36%) | Simvastatin: 50 (22%) |
| | Lamivudine: 15 (15%) | Tramadol: 45 (35%) | Tramadol: 45 (20%) |
| | Amlodipine:15 (15%) | Atenolol: 32 (25%) | Amlodipine: 44 (19%) |
| | Telmisartan: 12 (12%) | Amlodipine: 29 (22%) | Atenolol: 38 (17%) |
| | Sildenafil: 10 (10%) | Hydrochlorothiazide: 26 (20%) | Hydrochlorothiazide: 35 (15%) |
| | Efavirenz: 9 (9%) | Carvedilol: 22 (17%) | Spironolactone: 31 (14%) |
| | Hydrochlorothiazide: 9 (9%) | Metformin:17 (13%) | Carvedilol: 29 (13%) |
| | Zidovudine: 9 (9%) | | |

AF, atrial fibrillation; DCMO, dilated cardiomyopathy; DM, diabetes mellitus; HF, heart failure; HHD, hypertensive heart disease; HPT, hypertension; IHD, ischaemic heart disease; VHD, valvular heart disease; VTE, venous thrombo-embolism.

* More than one indication may exist; therefore, totals may exceed 100%.

**Other: included chronic venous insufficiency, antiphospholipid syndrome, thrombophilia, congenital heart block and thrombotic syndrome

Unfortunately, our findings complement other recent sub-Saharan studies in South Africa, Namibia and Botswana, all reporting poor INR control, with mean/median TTR ranging from 29% to 47% [1,12,13]. It is known that a higher TTR is associated with increased efficacy of warfarin and reduction in bleeding complications [14]. In patients with non-valvular AF receiving warfarin, there is a three-fold risk of ischaemic events with under-anticoagulation and a five-fold risk of bleeding events with over-anticoagulation, compared to being within the therapeutic range [14]. Maximising TTR is therefore crucial for attainment of good clinical outcomes, as shown by a study in the UK, where a 10% increase in time out of range was associated with a 29% increased risk of mortality, a 10% increased risk of ischemic stroke, and 12% increased risk of other thromboembolic events [15].

A number of factors may affect INR control. We found that INR testing was generally available at all sites; however, patients at the Ugandan clinics had to pay up to £4 for a laboratory INR test. In addition, frequent pharmacy stock outs of warfarin at these clinics means that patients also have to incur costs of privately purchasing warfarin. These costs may be prohibitive, particularly in the Ugandan population where the average household monthly income is

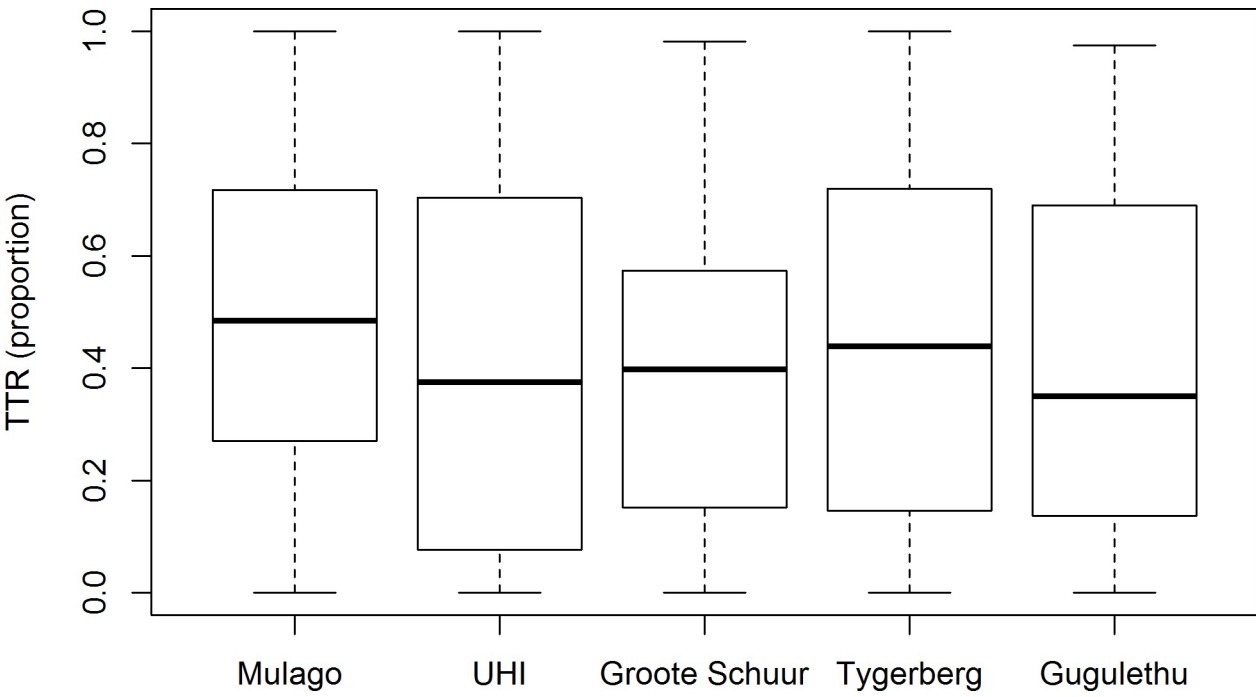

**Fig 1. Time in therapeutic INR range (TTR) of patients attending five anticoagulation clinics in Uganda and South Africa (box represents median and interquartile range).**

£62 - £144 [16]. However, TTR was poor at South African anticoagulation services as well, despite patients not having to pay for INR testing or warfarin.

Laboratory turnaround time at two clinics was at least one day requiring a second visit for dose adjustment, also contributing to increased transport costs and missed work days. Thus, strengthening laboratory facilities and pharmaceutical supply chain and provision of near-patient INR testing may contribute to improved anticoagulation control. However, TTR was poor even at facilities with onsite INR testing and no problems of drug stock outs, suggesting that there are other factors contributing to poor TTR.

Our population had a high burden of comorbidities such as heart failure and diabetes, which may be associated with lower TTR [17], because of polypharmacy as well as physiological changes due to comorbidity [18]. In addition, high HIV prevalence may further complicate warfarin dosing due to drug-drug interactions with non-nucleoside reverse transcriptase inhibitors and protease inhibitors [19,20]. Although our sample included few TB patients, it is important to consider that rifampicin, which is used in first-line TB treatment, induces warfarin metabolism, which further complicates warfarin dosing [21].

Pharmacogenomic differences due to polymorphisms in *VKORC1* and *CYP2C9* genes and additional polymorphisms in other genes e.g. *CALU* rs339097 may affect dosing requirements in patients of African descent [22]. Patients managed using genotype guided dosing algorithms in high-income settings have been shown to have higher mean TTR and fewer episodes of excessive anticoagulation compared to usual dosing [8]. Further studies need to be conducted in our population to develop and evaluate clinical and genetic dosing algorithms relevant for low-income settings.

DOACs require less frequent monitoring and are non-inferior or even superior to warfarin in preventing stroke in non-valvular AF [23,24]. However, their access is severely limited in

resource-limited settings due to cost limitations; necessitating urgent improvements in warfarin management.

This study has limitations. We only included urban sites, and findings may not be generalizable to rural settings in South Africa and Uganda. However, in Uganda, we included the clinics known to manage the majority of patients receiving warfarin anticoagulation. It is possible that there are additional challenges facing anticoagulation services in more remote and poorly resourced settings in both South Africa and Uganda, which we have not captured. We audited each included facility at a single timepoint.

In conclusion, in the South African and Ugandan clinics that we evaluated, TTR was suboptimal. Strategies to improve INR control in sub-Saharan patients taking warfarin are urgently needed as the alternative DOACs are not freely available. The findings from our work will inform further research to develop and validate a warfarin dosing algorithm, one such strategy which may improve TTR, for use in this setting.

## Supporting information

**S1 Appendix. Data collection form.**
(PDF)

**S1 Data.**
(XLSX)

## Acknowledgments

Thank you to the staff of Mulago National Referral hospital, Uganda Heart Institute, Groote Schuur Hospital, Tygerberg Hospital, and Gugulethu Community Health Centre for their help and support during the data collection for this work.

## Author Contributions

**Conceptualization:** Neil French, Mohammed Lamorde, Cheng-Hock Toh, Marc Blockman, Christine Sekaggya-Wiltshire, Catriona Waitt, Munir Pirmohamed, Karen Cohen.

**Data curation:** Andrea Jorgensen.

**Formal analysis:** Johannes P. Mouton, Andrea Jorgensen.

**Funding acquisition:** Neil French, Mohammed Lamorde, Cheng-Hock Toh, Marc Blockman, Christine Sekaggya-Wiltshire, Catriona Waitt, Munir Pirmohamed, Karen Cohen.

**Investigation:** Jerome Roy Semakula, Johannes P. Mouton, Shaazia Allie, Lynn Semakula.

**Project administration:** Claire Hutchinson.

**Supervision:** Christine Sekaggya-Wiltshire, Catriona Waitt, Munir Pirmohamed, Karen Cohen.

**Visualization:** Andrea Jorgensen.

**Writing – original draft:** Jerome Roy Semakula, Johannes P. Mouton.

**Writing – review & editing:** Jerome Roy Semakula, Johannes P. Mouton, Andrea Jorgensen, Claire Hutchinson, Shaazia Allie, Lynn Semakula, Neil French, Mohammed Lamorde, Cheng-Hock Toh, Marc Blockman, Christine Sekaggya-Wiltshire, Catriona Waitt, Munir Pirmohamed, Karen Cohen.

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
