## [Decision Letter · Decision Letter 0]

8 Oct 2019

PONE-D-19-22709

A cross-sectional evaluation of warfarin anticoagulation services in Uganda and South Africa

PLOS ONE

Dear Dr. Cohen,

Thank you for submitting your manuscript to PLOS ONE. After careful consideration, we feel that it has merit but does not fully meet PLOS ONE’s publication criteria as it currently stands. Therefore, we invite you to submit a revised version of the manuscript that addresses the points raised during the review process.

We would appreciate receiving your revised manuscript by Nov 22 2019 11:59PM. To enhance the reproducibility of your results, we recommend that if applicable you deposit your laboratory protocols in protocols.io, where a protocol can be assigned its own identifier (DOI) such that it can be cited independently in the future. For instructions see: http://journals.plos.org/plosone/s/submission-guidelines#loc-laboratory-protocols

We look forward to receiving your revised manuscript.

Kind regards,

Joel Msafiri Francis, MD, MS, PhD

Academic Editor

PLOS ONE

Journal Requirements:

2. Please include additional information regarding the survey or questionnaire used in the study and ensure that you have provided sufficient details that others could replicate the analyses. For instance, if you developed a questionnaire as part of this study and it is not under a copyright more restrictive than CC-BY, please include a copy, in both the original language and English, as Supporting Information. Moreover, please include more details on how the questionnaire was pre-tested, and whether it was validated.

Additional Editor Comments (if provided):

Reviewers' comments:

Reviewer's Responses to Questions

**Comments to the Author**

1. Is the manuscript technically sound, and do the data support the conclusions?

Reviewer #1: Yes

Reviewer #2: Yes

Reviewer #3: Partly

2. Has the statistical analysis been performed appropriately and rigorously? 

Reviewer #1: Yes

Reviewer #2: Yes

Reviewer #3: No

3. Have the authors made all data underlying the findings in their manuscript fully available?

Reviewer #1: Yes

Reviewer #2: Yes

Reviewer #3: No

4. Is the manuscript presented in an intelligible fashion and written in standard English?

Reviewer #1: Yes

Reviewer #2: Yes

Reviewer #3: Yes

5. Review Comments to the Author

Reviewer #1: SUBMISSION SUMMARY:

Semakula and colleagues present an interesting and important cross sectional study examining the demographics and time in therapeutic range (TTR) of patients receiving warfarin at five anticoagulation clinics in Uganda and South Africa. In particular, they report on facility characteristics, indications for anticoagulation, patient characteristics and comorbidities, and anticoagulation care quality metrics.

GENERAL IMPRESSIONS:

Overall, the investigators explore an important clinical question, namely the indications for warfarin as well as TTR attainment in several anticoagulation clinics in Sub Saharan Africa. The manuscript could be strengthened, however, by further clarification and enumeration of the methods, both in the rationale for the timeframe and patient selection procedure as well as statistical methods. Furthermore, since individual level data were gathered from the chart review, multivariable regression analyses could be performed to determine which variables may be associated with better TTR, thus exploring many of the speculative mechanisms for poor treatment quality brought up by the authors in the Discussion section.

COMMENTS BY SECTION:

* Title:

- Consider clarifying the title to avoid misunderstanding that the evaluation was performed on a nationwide scale (e.g. “A cross-sectional evaluation of warfarin anticoagulation services in five clinics in Uganda and South Africa”).

* Abstract:

- Page 3 (Line 55): Consider using more specific language than “INR control”. This is acceptable clinical wording, but the conclusion suggested by this manuscript specifies inadequate TTR attainment is the finding, which may be a more precise description of the reported outcome.

* Introduction:

- Page 4 (Line 68): Citation needed.

- Page 4 (Line 70): Citation needed.

* Methods:

- Page 5 (Line 81): Was there a reason that a 7 month data collection period was selected? It seems that 12 months would be a more relevant/representative timeframe, particularly since there are seasonal variations in hospitalization rates and causes, which may impact the generalizability of the findings. If there was not a specific reason for the selected time interval, this should be reported as a limitation of the study.

- Page 5 (Line 93): Please describe the rationale and specific procedure for the convenience sample. Were patients enrolled consecutively? Were they enrolled following a clinic visit? Was there any random selection procedure of charts? Enrollment alphabetically could potentially introduce complicating factors such as overrepresentation of certain families, ethnic groups, etc.

- Page 5 (Line 94): If a hypothesis of this paper is to assess for potential medication interactions between warfarin and TB/HIV drugs, perhaps it would make more sense to select and query the percentage of patients taking medications known a priori to affect warfarin metabolism. Examples of such medications (NNRTIs and rifampicin) are brought up by the authors in the Discussion section (Page 11) but are not reported as included among drugs queried in the study design.

- Page 5 (Line 96): “Statistical and data analysis” is redundant. The subheading can be changed to “Statistical Analysis” or “Data Analysis”.

- Page 5 (Line 98): Please specify which summary statistics were used (e.g. median, IQR, etc.).

- Page 5 (Line 99): Please name what software packages were used for statistical analysis and figure generation.

- Page 5 (Line 99): This study could be strengthened by performing regression analyses, which could potentially identify factors that may be associated with substandard TTR attainment. Was there a reason this was unable to be performed?

- Page 6 (Line 104): Please include the basis for waiving of informed consent from patients.

- Page 6 (Line 109): Please include the basis for waiving of informed consent from staff.

* Results:

- Page 6 (Lines 111-117): The descriptions of the audited clinics and interviewed individuals could be included in the Study Design and Sites subsection rather than the Results section, since these were known prior to the acquisition of study data.

- Page 8 (Line 133, Line 136): Please clarify the analyses were performed in “Ugandan sites” and “South African sites” rather than “Uganda” or “South Africa”.

* Discussion:

- Page 10 (Line 166): Please clarify how regular patient follow-up and regular INR monitoring were confirmed in this present analysis (e.g. what percentage of patients in the program actually made it to monthly follow-up appointments or INR monitoring visits).

- Page 11 (Lines 189-193): See above in Methods section. The authors discuss the potential of NNRTIs or rifampicin to affect warfarin metabolism, but the prevalence of use of these drugs is not explicitly queried in the study design.

Reviewer #2: 1. Authors state that they used a convenience sample. What was this convenience? Was it consecutive patients seen in the clinics? Random? Needs some elaboration.

2. The authors stated that they interviewed service providers (Doctors, nurses, etc.). However that data obtained from these interviews (quantitative or qualitative) are not clearly mentioned or discussed in the results or discussion sections. Most of the data they state in results section could have come from the patients and patient records. This needs to be clarified.

Reviewer #3: Reviewer # comments

Manuscript title: A cross-sectional evaluation of warfarin anticoagulation services in Uganda and South Africa

General feedback: The paper addresses an important topic on warfarin in the lower-income country (Uganda) and middle-income country (South Africa)

However, several methodological issues need to be addressed.

1. Methods:

a. The audit was conducted between 1 January and 31 July 2018. Authors extracted demographic data and clinical data including indication for anticoagulation, comorbidities, concomitant medications and INR results. It would have been essential to state the span over which INR results were collected. Did the authors document all the INR results over the study period (1 January and 31 July 2018)?

b. We selected, through convenience sampling, key members of clinic staff who were involved in anticoagulation care and who indicated their availability and willingness to be interviewed. Interviews were conducted using a structured questionnaire. I would avoid the word interview but rather use questionnaire administration. This was not a qualitative study, but rather the administration of structured questionnaires.

2. Statistical and data analysis

a. Although the authors stated that time in therapeutic range (TTR) over the most recent

three months was calculated using the Rosendaal interpolation method, one would expect more description as to who qualified for analysis. The Rosendaal method uses INR values from patients with at least two valid intervals separated by 56 days (8 weeks) or less, without an intervening hospitalisation. Some authors have used 60-90 days between readings. It is, therefore, essential to state the interval.

3. Results:

a. A lot of the initial sentences in the results section fits more on the method section than as results. For instance, the description of the study setting. Knowing that Mulago is a tertiary hospital or Groote Schuur as a secondary hospital is a piece of basic information that describes a study site, even before data collection. Kindly review this part.

b. Kindly state if valvular heart diseases included mechanical heart valves.

4. Discussion

a. Although the authors pointed out that the cost of INR testing and warfarin is may be prohibitive in anticoagulation control, this was not clear in figure 1. Ugandan site where patients paid for the warfarin/INR service had a similar level of anticoagulation control as the South African site! I expected this observation to feature in the discussion than the way the authors discussed.

b. Similarly, the contribution of HIV was not very evident despite the considerable difference in the prevalence (20% vs 5%). As in (a) above, this was not clearly stated.

c. Although this study is typically descriptive and may not be powered enough to analyse associations, it would have been nice to look at whether the variables which were collected had any influence on the level of anticoagulation control. If this is not done, the discussion on ‘’associated factors’’ will be limited.

d. Study limitations. How about the sample size? Was it enough to answer the primary objective? This is not justified in the manuscript and could feature as a limitation.

e. Conclusion: The findings from our work will inform further research to develop and validate a warfarin dosing algorithm for use in this setting. Was there any evidence to suggest that the absence of a validated algorithm affected anticoagulation control? As some sites had adopted some algorithm, one would expect the authors to have analysed if the lack of an algorithm affected anticoagulation control before having this conclusion.

6. PLOS authors have the option to publish the peer review history of their article (what does this mean?). If published, this will include your full peer review and any attached files.

Reviewer #1: No

Reviewer #2: No

Reviewer #3: Yes: Julius Mwita

---

## [Author Response · Author response to Decision Letter 0]

25 Nov 2019

Please see the uploaded file, in which we respond to all comments.

---

## [Decision Letter · Decision Letter 1]

4 Dec 2019

PONE-D-19-22709R1

A cross-sectional evaluation of five warfarin anticoagulation services in Uganda and South Africa

PLOS ONE

Dear Dr. Cohen,

Thank you for submitting your manuscript to PLOS ONE. After careful consideration, we feel that it has merit but does not fully meet PLOS ONE’s publication criteria as it currently stands. Therefore, we invite you to submit a revised version of the manuscript that addresses the points raised during the review process.

Specifically, 

There are two unclear issues – that you may need to rectify before the paper is accepted for publication.

Why no formal sample size estimation was carried out? This is critical and justification is important.I am not sure it is true that there was no sampling frame?  Ideally – the sampling frame should have been the number of patient records available at the sites.  So, the statement that there was no sampling frame may not be entirely true.

We would appreciate receiving your revised manuscript by Jan 18 2020 11:59PM. To enhance the reproducibility of your results, we recommend that if applicable you deposit your laboratory protocols in protocols.io, where a protocol can be assigned its own identifier (DOI) such that it can be cited independently in the future. For instructions see: http://journals.plos.org/plosone/s/submission-guidelines#loc-laboratory-protocols

We look forward to receiving your revised manuscript.

Kind regards,

Joel Msafiri Francis, MD, MS, PhD

Academic Editor

PLOS ONE

Reviewers' comments:

Reviewer's Responses to Questions

**Comments to the Author**

1. If the authors have adequately addressed your comments raised in a previous round of review and you feel that this manuscript is now acceptable for publication, you may indicate that here to bypass the “Comments to the Author” section, enter your conflict of interest statement in the “Confidential to Editor” section, and submit your "Accept" recommendation.

Reviewer #2: All comments have been addressed

Reviewer #3: All comments have been addressed

2. Is the manuscript technically sound, and do the data support the conclusions?

Reviewer #2: Yes

Reviewer #3: Yes

3. Has the statistical analysis been performed appropriately and rigorously? 

Reviewer #2: Yes

Reviewer #3: Yes

4. Have the authors made all data underlying the findings in their manuscript fully available?

Reviewer #2: Yes

Reviewer #3: Yes

5. Is the manuscript presented in an intelligible fashion and written in standard English?

Reviewer #2: Yes

Reviewer #3: Yes

6. Review Comments to the Author

Reviewer #2: My comments have been fully addressed. I don't have further comments. I think the manuscript adds to the existing knowledge.

Reviewer #3: comments have been addressed.

However, authors are advised to check the whole manuscript for consistency of their wordings. For instance the use of "INR control" continued despite of the fact that authors agreed to replace it. Similarly, the word "interview" still appears in the abstract

7. PLOS authors have the option to publish the peer review history of their article (what does this mean?). If published, this will include your full peer review and any attached files.

Reviewer #2: No

Reviewer #3: No

---

## [Author Response · Author response to Decision Letter 1]

10 Dec 2019

Please refer to the uploaded document for a point-by-point response to the editor's and reviewer's comments.

---

## [Editor Report · Decision Letter 2]

19 Dec 2019

A cross-sectional evaluation of five warfarin anticoagulation services in Uganda and South Africa

PONE-D-19-22709R2

Dear Dr. Cohen,

We are pleased to inform you that your manuscript has been judged scientifically suitable for publication and will be formally accepted for publication once it complies with all outstanding technical requirements.

With kind regards,

Joel Msafiri Francis, MD, MS, PhD

Academic Editor

PLOS ONE
---

## [Editor Report · Acceptance letter]

17 Jan 2020

PONE-D-19-22709R2 

A cross-sectional evaluation of five warfarin anticoagulation services in Uganda and South Africa 

Dear Dr. Cohen:

I am pleased to inform you that your manuscript has been deemed suitable for publication in PLOS ONE. Congratulations! Your manuscript is now with our production department. 

With kind regards,

on behalf of

Dr. Joel Msafiri Francis 

Academic Editor

PLOS ONE